# Direct Solar Oven with and without UV Filter vs. Traditional Oven: Effect on Polyphenolic Antioxidants, Vitamins and Carotenoids of Food

**DOI:** 10.3390/molecules28083519

**Published:** 2023-04-17

**Authors:** Seyed Sepehr Moeini, Chiara Dal Bosco, Elena Mattoni, Tecla Gasperi, Alessandra Gentili, Daniela Tofani

**Affiliations:** 1Department of Science, “Roma Tre” University, Via della Vasca Navale 79, 00146 Rome, Italy; seyedsepehr.moeini@uniroma3.it (S.S.M.); elena.mattoni@uniroma3.it (E.M.);; 2Department of Chemistry, “Sapienza” University, Piazzale A. Moro, 00185 Rome, Italy; chiara.dalbosco@uniroma1.it (C.D.B.); alessandra.gentili@uniroma1.it (A.G.)

**Keywords:** polyphenols, vitamins, vitamers, antioxidants, solar cooking

## Abstract

The use of efficient solar ovens can be a way for progressing ecofriendly technologies in the field of food preparation. In most performing direct solar ovens, the sun rays reach the foods directly, therefore, it becomes essential to check whether, in these conditions, foods can retain theirnutraceutical properties (antioxidants, vitamins and carotenoids). In the present research work, to investigate this issue, several foods (vegetables, meats, and a fish sample) were analyzed before and after cooking (traditional oven, solar oven, and solar oven with a UV filter). The content of lipophilic vitamins and carotenoids (analyzed via HPLC-MS) and the variation of total phenolic content (TPC), and antioxidant capacity (via Folin–Ciocalteu and DPPH assays) have evidenced that cooking with the direct solar oven allows to preserve some nutrients (i.e., tocopherols) and, sometimes, to enhance nutraceutical properties of vegetables (for example, solar oven-cooked eggplants showed 38% higher TPC compared to electric oven cooked sample) and meats. The specific isomerization of all-trans-β-carotene to 9-cis was also detected. The use of a UV filter is advisable to avoid UV drawbacks (for instance, a significant carotenoid degradation) without losing the beneficial effects of the other radiations.

## 1. Introduction

In recent decades the need for renewable energy has become a compelling ecological demand that affects all human activities [1]. Recently, in the food and agriculture industry, solar energy has been used as energy supply to lower the costs, to dry foods (solar dryer) or applied to waste transformations [2,3,4]. The use of solar cookers was initially developed to help the population during calamities or to diminish the people’s need of wood or charcoal for water sterilization and food preparation [5,6]. Recently, the new ecological sensitivity of modern society has led some people to extend the use of solar cookers in natural environments, such as camping areas or outdoor excursions. The simplest models of solar cookers use cardboard parabolic reflectors to focalize the sun rays on a covered black pot, generally put inside a plastic bag to insulate. The heat introduced permits cooking at medium temperature (65–150 °C) and the food is generally stewed in water, sauce, or oil.

To encourage the widespread use, a solar cooker should have the same performance of a traditional electric oven or grill, both in the achievable temperature and in the possible types of cooking (roast, stewed and grilled). To obtain these goals, id est high heating efficiency and good insulation, one satisfactory solution could be the use of a “direct solar oven” in which the sun rays reach the food directly in an insulated chamber.

On the other hand, sun rays cause the food to interact with a broader radiation spectrum, ranging from UV to IR. This can deteriorate the more unstable edible molecules, such as lipids, carotenoids, and antioxidants [7]. However, a limited number of studies have analyzed the influence of direct sun exposure on the nutritional parameters of cooked food.

In the present study, the influence of direct solar cooking is investigated on a number of nutritional parameters of the foods. Vitamins, total phenolic content (TPC), and antioxidant capacity of a series of food materials, either raw or cooked with electric oven, or with direct solar oven were determined. Furthermore, to check the effect of the more energetic UV radiations, analyses were also performed on food cooked using a solar oven equipped with an UV plastic filter that blocked all the radiation lower than 400 nm. All the data were compared with each other, and some promising results are presented.

## 2. Results

### 2.1. Oven, Choice of Food Sampling, Cooking and Sample Preparation for the Analyses

In this research, the Helio© solar cooker was used as a direct solar oven (Figure 1, see Material and Methods for the technical features), to cook the chosen food.

Different kinds of food were chosen for the analyses: carrots, eggplants, onions, peppers, and zucchini as vegetables; pork loin, chicken breast as meat and cod fish filet as fish samples. In the vegetable samples, high content of antioxidants, either polyphenolics or carotenoids, are generally present. Furthermore, these kinds of vegetables can be baked or grilled and the effect of solar cooking should be greater if direct sun rays are used in the process without the presence of water (boiling or steaming). The meat and fish were chosen to check if the “Maillard reaction” could change the antioxidant composition even in higher protein food. To avoid problems associated with freshness, frozen fish was preferred. In this case, it was defrosted before cooking.

Each food was cooked at the same temperature in all the devices until complete baking (see Appendix A).

In case of vitamins and carotenoids analysis, cold saponification (CS) was applied to hydrolyze fatty acids esterified with carotenols, with the dual advantage of simplifying the interpretation of the chromatographic profiles and obtaining greater enrichment factors useful for the detection of minor carotenoids (see Section 4.2).

In the determination of the total phenolic content and the 2,2-diphenyl-1-picrylhydrazyl radical (DPPH) assay, samples for analysis were obtained after solvent extraction (see Section 4.3 for details).

### 2.2. Vitamins and Carotenoids Analysis

High performance liquid chromatography (HPLC) coupled with diode array (DAD) and mass spectrometry detection (MS) was used to analyze vitamins and carotenoids. As most carotenoids have a characteristic three-peaks spectrum responsible for strong absorptions in the UV-Visible region of the electromagnetic spectrum, the identification of the cis β-carotene isomers was based on the comparison between their UV spectrum and that of the corresponding all-trans isomer. Such an analytical strategy was applied to study how the content of fat-soluble micronutrients is altered after cooking food samples in the conventional electric oven, in the Helio solar oven and in the solar oven with UV filter [8].

Qualitative and semi-quantitative (mean chromatographic areas) analyses were performed on raw and cooked foodstuff. Table 1 reports an overview of the fat-soluble micronutrients detected.

Among raw vegetables, in the case of peppers, zucchini and eggplants, at least 80% of the fat-soluble micronutrients is represented by vitamin E, with a clear prevalence of α-tocopherol, while the complementary percentage includes several carotenoids, mainly phytoene and β-carotene, or lutein in the case of zucchini.

This picture is inverted for carrots, with a content of carotenoids as high as 68%, almost equally represented by β-carotene, phytoene and phytofluene, while the remaining 32% is vitamin E (mainly α-tocopherol).

In the literature, the β-carotene content varies between 200 μg/g in eggplants, 2200 μg/g in zucchini and up to 59,120 μg/g in carrots. The raw samples of these three vegetables analyzed in this work showed a percentage content of β-carotene comparable to that in the literature (β-carotene content of 0.37% in eggplant and 3.7% in zucchini in comparison with 0.34% and 3.1%, respectively, in the literature) [9].

In meat and fish samples (chicken, pork loin and cod fish) no fat-soluble vitamins nor carotenoids were detected, with the exception of chicken, which had a relative composition in fat-soluble micronutrients, 85% vitamins A and E (with 59% retinol and 22% γ-tocopherol isomers, respectively), and 15% carotenoids (mainly lutein).

When considering the effect of cooking, it can be seen that the chromatographic areas of E vitamers are almost unchanged in the raw and cooked foodstuff. On the contrary, with the exception of phytoene, the content of carotenoids is reduced by about 50% by the temperature (electric and UV filtered solar ovens), and even more by the combined effect of temperature and UV light (solar oven). In Figure 2, the data obtained on raw carrot are reported as an example.

One further effect of cooking is the conversion of all-transβ-carotene into the cis-forms. Insic et al. reported that raw carrots contain 62.28% of all-trans β-carotene in comparison with the other isomers, while the boiling lowers this value to 59.18% and enhances that of 13-cis from 0.55% to 5.75% [7]. In this research, raw carrots contained 78.1% of all-trans isomer, with 13.3% of 13-cis and 8.5% of 9-cis. The electric oven favored the formation of the 13-cis (32.1%) rather than the 9-cis (8.3%) in comparison with the all-trans isomer (59%). On the other hand, the solar oven led to an almost equivalent increase in both forms (13-cis to 34.4% and 9-cis to 21.6%), which were not significantly lowered by the use of UV filter (13-cis to 27.5% and 9-cis to 25.1) (Figure 3).

### 2.3. Total Phenolic Content (TPC)

The total phenolic content was measured using the Folin–Ciocalteu assay, a spectrophotometric method which allows the quantification of the phenolic and polyphenolic molecules responsible for most of the antioxidant activity, especially in vegetables [10]. The measures of TPC content, either in raw or cooked food, were expressed as gallic acid equivalents (GAE), i.e., the mg of gallic acid corresponding to the polyphenols present in l g of dry food material. Data are shown in Table 2.

In raw vegetables the total phenolic content varies from 19 mg GAE/g DW in carrots cooked in the electric oven to 207 mg GAE/g DW in eggplants cooked in the solar oven. Eggplants, peppers, and zucchini showed higher TPC values, either raw or cooked. For example, a 38% increase in TPC from electric oven to solar oven cooking was observed in eggplant samples while the increase was 24.6% from electric oven to filtered solar oven cooking. It is difficult to compare the obtained data with the literature because of the variability of cultivar (see peppers, eggplants, or zucchini), cultivation conditions and harvest time. Furthermore, TPC can be calculate as percentage or as GAE on the basis of fresh or dry material weight. In the case of raw carrots, TPC is coherent with Lutz data (19.9 ± 1.1 mgGAE/g) [11]. The analyzed red sweet peppers showed higher TPC than the red almuden cultivar (62.6 ± 1.1 mg GAE/g) [11] or the serrano (10.32 ±0.95 mg GAE/g DW) [12]. Analogously, various eggplant cultivars showed a TPC that ranges from 615 to 1376 mg/kg [13].

However, the main aim of this work is the comparison between raw and cooked samples to analyze the effect of different cooking methods free from other variables. After cooking, TPC values changed and the increase in the phenolic content was significant in cooked carrots, eggplants, and peppers. This can be attributed to the release of polyphenols from the proteins where they are bound [14,15].

Arkoub et al. confirmed that, in eggplants, the cooking process varies the TPC from 48.38 ± 6.7 mg GAE /g DW in the raw material to 77.93 ± 2.18 mg GAE /g DW in the baked sample [16]. These data are both lower than those presented here but in line with our findings.

In meat and fish, the total phenolic contents were lower than in vegetables, except in chicken samples (47 ± 3 mg GAE/g DW) that contained three times the polyphenols of pork loin and twice of cod fish. In these cases, after cooking, the difference of TPC with raw samples was lower (cod fish) or under the limit of the error (chicken and pork loin). Literature data showed different TPCs in pig meat depending on genetic type with average values of 128 mg GAE/100 g and 75 mg GAE/100 g in raw and cooked pig meat, respectively [17]. Additionally, feeding (130–300 mg GAE/100 g FW) and sample location in the carcass can further affect the TPC [18].

The difference of phenolic content in food cooked with the direct solar oven can be attributed to the effects of UV rays that could further enhance cell degradation and ease the antioxidants release [19]. This is confirmed by the significant decrease in the TPCs that were measured in eggplants and peppers when the solar oven was used with the UV filter.

Where the effect of the solar oven is less evident, either without or with the UV filter, it can be supposed that there was a balance between the release of polyphenols and their degradation.

### 2.4. DPPH Assay

The standardized alcoholic extracts were also analyzed in their antioxidant capacity using DPPH assay [20]. This method, based on the electron transfer capacity of the antioxidant, is widely used to test the antioxidant capacity of natural compounds and foods, either raw or cooked [21]. Data of all DPPH analyses are presented in Table 3 as EC50 *id est* the effective concentration of sample that reduces the absorbance of a standard solution of the DPPH radical by 50%. The EC50 values proved to be lower in samples with higher free radical scavenging ability.

The EC50s of raw vegetables were generally lower than those of meat and fish as in TPC analyses. This is in line with the literature data that evidence a higher antioxidant capacity of eggplants (543.3 ± 12.5 umol TE/gFW) respect to red peppers (449.8 ±10 umol TE/gFW) or carrots (303.3 ± 7.6 umol TE/gFW) [11]. Data on green zucchini evidence 38.5% radical scavenging ability, half that of ascorbic acid in the same conditions [22], but can hardly be compared with the data obtained in the present study.

In the case of meats, the comparison of data with the literature is not simple. Many articles dealt with antioxidant activity of meat derived by particular animal diet supplementation (amino acids or specific vegetables) to enhance the food stability or nutritional properties. For instance, the DPPH of “control chicken” breast showed a value of 13.5 mM Trolox/g FW [23], while the Celta Pigs fed with commercial products showed a value of 957.33 ± 101.50 μg TE/g FW [18]. As for cod fish, many studies dealt with the antioxidant activity of its protein hydrolysate [24] but no DPPH analysis of fish muscles were reported.

However, the comparison of the data obtained on raw and cooked food evidence that the EC50 of all foods were higher in raw samples than in cooked products, confirming that cooking increased the antioxidant capacity due to a higher polyphenol release and an enhancement of the antioxidants produced by Maillard reaction.

Examining the different baking techniques used, food samples showed comparable or higher (pepper, zucchini, chicken, and cod fish) antioxidant capacity in products of solar cooking with respect to traditional cooking. It could be supposed that the UV rays accelerated protein degradation, leading to further release of antioxidants (see Section 2.3). Only carrots showed a similar antioxidant activity in the sample cooked with solar oven and the traditional one. This could be explained by assuming a balance between the carotenoids loss, β-carotene isomerization (see Section 2.2) and the higher concentration of free polyphenols (Section 2.3).

When the solar oven was used with the UV filter, the antioxidant capacity was slightly enhanced (carrots, zucchini, and pork loin) or remained similar to that of solar baking. The higher radical scavenging capacity of carrots cooked with the filtered solar oven can be explained by the lower degradation of β-carotene when the UV rays shield was used.

## 3. Discussion

Together with higher digestibility, the cooking of vegetables and meats increases the free polyphenols and carotenoids of food. This is coherent with the literature data that underlines how the cooking process produces two main effects: (i) in vegetables, the antioxidants, present in pectin or cellulose, are partially or totally released and these become more bioavailable [25,26]; (ii) many Maillard products are themselves antioxidants and enhance the radical scavenging activity of the foods [27].

This phenomenon is enhanced by UV rays, producing higher content of antioxidants in solar cooked food, and it is particularly evident in vegetables with plenty of polyphenols (eggplants and peppers, Section 2.3). However, UV radiation and heat can also degrade some vitamins, carotenoids, or polyphenols.

Fat-soluble vitamins and carotenoids are a large family of essential micronutrients having a variety of functions crucial for the human organism. The fat-soluble vitamins (A, D, E and K), are present as a large number of homologues, called vitamers. Vitamers have a different bioavailability, biological potency, and physiological role [28,29]. The elucidation of food vitamers and carotenoids composition is a challenging task because of their instability under light, oxygen, and pH extremes. Other analytical difficulties lie in separating a large number of compounds and in distinguishing structural and geometrical isomers. Currently, most methods have only dealt with the analysis of one or few major forms of vitamins and carotenoids. Procedures like these might lead to oversimplified results because the natural distribution of the minor, but not less important vitamin and carotenoid forms, remains unrevealed. This could prevent finding a correlation with nutritional properties of foods [30]. For this reason, in this research, HPLC, coupled with diode array (DAD) and mass spectrometry detection (MS), was used to increase sensitivity, selectivity, and identifiability of vitamins and carotenoids.

With regard to these lipophilic compounds (Section 2.2), analyses underlined that the cooking does not affect the content of vitamin E. The thermal degradation of tocopherols depends on several factors, including temperature, heating time, food matrix and cooking method [31]. Under the applied experimental conditions, all ovens led to a vitamin loss within the experimental error (20%), and therefore to statistically insignificant differences between samples (see Table 1). As far as carotenoids are concerned, their loss is a function of the number of conjugated double bonds in their structure and becomes significant as the extent of conjugation increases. In practice, only phytoene, which contains three conjugated double bonds, does not undergo thermal degradation, while the loss of β-carotene, which has eleven conjugated double bonds, is approximately 50%. Such loss is consistent with data reported in the literature for similar temperatures and heating times [32]. Moreover, carotenoids are known to be prone to photooxidation [33], therefore, their loss is maximum in food cooked with the solar oven. The positive effect of the UV filter used with the Helio apparatus is evident from the obtained results (see also Figure 2), which showed carotenoids contents comparable to those resulting from baking in the electric oven and higher in comparison with those of solar cooking.

The isomerization of β-carotene is also increased by solar cooking (Figure 3). Additionally, in this case, the increase in cis isomers percentage between electric (sum of cis isomers 41%) and solar cooking (all cis isomers 66%) appears to be positively affected by the use of the UV filter. As a matter of fact, using only the visible radiation, the percentage of 13-cis isomer decreased to 27.5% from the 34.4% in solar cooking, even lower than electric cooking (32.1%), but a high isomerization of β-carotene to the 9-cis isomer (25.1% respect to 8.3% in electric) was maintained.

In contrast with the data on carotenoids, the TPC of raw samples showed a general increase in polyphenols in cooked food, particularly evident when vegetables were cooked with the solar oven. The TPC, after solar cooking, increased by 41.7% in the eggplant compared to raw sample, 13.2% in the peppers, 10% in the carrots, and 5.8% in the zucchini (Table 2). This might be explained considering that the higher energy of UV radiation could enhance the Maillard reaction and/or increase the release of the polyphenols from biological macromolecules [25].

Concerning the antioxidant capacity of cooked foods, this is the first time that the combined effects of UV and heating have been analyzed. The DPPH assay data are derived from all the substances that participate in radical scavenging, either polyphenolics or vitamins. Considering the lowering of carotenoids and xanthophylls caused by heat and UV rays, it can be assumed that in cooked samples, the scavenging effect could be mainly attributed to the tocopherols, which are more stable under heat, and to the higher quantity of polyphenols released from proteins and macromolecules during the cooking process. In some foods (onions, eggplants, pork loin and codfish) the two effects balance each other, so results show that the solar oven does not change the radical scavenging ability of food if compared with traditional cooking techniques (Table 3). Only carrots presented EC50 of the sample slightly higher than the electric one, probably due to the higher carotenoids loss and β-carotene isomerization that produced isomers which could be less antioxidant than the natural products [34]. On the other hand, some foods (pepper, zucchini, and chicken) showed higher antioxidant capacity after solar cooking than compared to traditional oven baking.

In some cases, when the solar oven was used with the UV filter, the antioxidant capacity further increased (carrots and pork loin). This can be explained considering that when the filter is used, the vitamin contents rise back to the levels of a traditional baking and this can further enhance the antioxidant capacity.

## 4. Materials and Methods

All reagents and standards were purchased from Sigma-Aldrich (Merk group, Darmstadt, Germany). Solvents used for extraction and analyses were HPLC grade.

Carrots (*Daucus carota, L. cv. sativa, D.C*.), eggplants (*Solanum melongena, L*.), onions (*Allium cepa, L*.), red sweet peppers (*Capsium annuum, L., Nocera cultivar*), and zucchini (*Cucurbita pepo, L. Milano black*), were purchased fresh from a general supermarket. Chicken breasts (*Gallus gallus domesticus*) and pork loins (*Sus scropha domesticus*) were purchased fresh, already cut into thin slices. Frozen cod fish (*Gadus morhua*) filets were purchased and defrosted before processing. At least four different samples of all vegetables or four different slices of meat or fish were taken.

### 4.1. Helio© Solar Cooker

This solar oven was used for cooking food samples. It has eight reflectors made of light wood panels covered with reflecting material and positioned as a parabolic mirror that directs the sun rays into the cooking chamber through the glass. The cooking chamber can reach up to 300 °C in 20–30 min during a sunny day and maintain the temperature until the sun rays fade. Two thermometers were used to monitor the temperature of the internal air and of the pan. To lower the temperature, the small rear door of the oven was kept slightly open until the proper temperature was acquired and maintained in a range of ±10 °C. In the cooking filter, a special plastic UV filter was placed onto the glass in order to shield food from rays with wavelengths < 400 nm.

### 4.2. Food Preparation

All food pieces were fresh and without any imperfection or damage. Each vegetable was washed with tap water, dried with paper, and sliced into pieces of uniform thickness. For each food material, four homogeneous samples were prepared by putting together pieces, with approximately the same weight and size, from different parts of the material. One sample was directly dried and, after the complete removal of water, frozen at −20 °C without cooking (raw sample, R) and stored until analyses.

The other samples were cooked in an electric oven (E), in the Helio solar oven (S), and in the solar oven equipped with the UV filter (F). The cooking was carried out at the same temperature in all the apparatus (see Appendix A) until the food was completely cooked. After cooking, all samples were weighted, and dry lyophilized (Telstar Lyoquest 55, Azbil Telstar Technologies—Barcelona, Spain) for days until a constant weight was achieved. Dry samples were divided, kept at −20 °C and separately prepared for each analysis.

### 4.3. Vitamins and Carotenoids Analysis

Before LC-MS analysis, dried food samples were subjected to an extraction protocol based on cold saponification followed by liquid–liquid extraction with hexane, as described in our previous work [35]. Briefly, freeze-dried samples were ground and 0.5 g aliquots were transferred into 50 mL polypropylene centrifuge tubes. After the addition of 6 mL absolute ethanol, containing 0.1% (*w/v*) BHT, and 1 mL of 50% (*w/v*) aqueous KOH, the samples were left under magnetic stirring overnight inside a water bath at 25 °C. The day after, the digest was diluted with 3 mL of ultrapure water and the analytes extracted with 4 mL of hexane, containing 0.1% (*w/v*) BHT. The extraction was repeated six times and after the addition of each fraction of solvent, the sample was stirred for 5 min, vortexed for 5 min and centrifuged at 6000 rpm at 4 °C for 10 min. Finally, the collected hexane fractions were neutralized by washing twice with 8 mL aliquots of ultrapure water, transferred into a glass tube, and evaporated at 30 °C, under a nitrogen flow, up to 100 µL. The extract was diluted to a final volume of 200 µL by adding a 2-propanol:hexane (75:25, *v/v*) solution containing 0.1% (*w/v*) BHT. Finally, 40 µL was injected into the HPLC-APCI-MS/MS system. Non-aqueous reversed-phase (NARP) conditions were used to separate analytes on a ProntoSIL C30 column (4.6 × 250 mm, 3 μm) (Bischoff Chromatography, Leonberg, Germany) chilled at 19 °C. The composition of the mobile phases, methanol (phase A) and 2-propanol/hexane (50:50, *v*/*v*; phase B), was varied according to the following gradient: t_0−1 min_, 0% B; t_1−15 min_, 0−75% B; t_15−15.1 min_, 75−99.5% B; and t_15.1−30.1 min_, 99.5% B. The mobile phase was completely introduced into the APCI source of the MS detector at a flow rate of 1 mL/min. The analytes were detected in positive ion mode by using the following conditions: the needle current at 3 μA; the probe temperature at 450 °C; the curtain gas was high-purity nitrogen at 40 psi; the collision gas was nitrogen at 4 mTorr; the nebulizer gas was air at 55 psi; and the makeup gas was air at 30 psi.

### 4.4. Sample Extraction for DPPH and Folin Analyses

Each lyophilized sample was smashed in a mortar. The weighted powder (generally 1 g) was extracted three times in ethanol using an ultrasound apparatus (Sonica Soltec 2002 MH) for 30′ at room temperature and then centrifuged (IEC CL31 Multispeed Centrifuge Thermo Scientific, Waltham, MA, USA) at 9000 rpm for 10 min. The procedure was repeated three times with methanol. All the solutions were collected together, evaporated under reduced pressure in a 30 °C bath rotavapor and then diluted in a 10 mL volumetric flask with methanol.

### 4.5. Total Phenolic Content Determination (TPC)

Food extracts were analyzed by using Folin–Ciocalteu reagent and the protocol already published [36], to assess the TPC. In brief, Folin–Ciocalteu commercial reagent was diluted (1:10) with deionized water (solution A). Each sample (0.100 mL) was mixed with 0.75 mL of this solution and allowed to stand for 3 min at 25 °C. Then, 0.75 mL of a saturated sodium carbonate solution was added. After 90 min, the absorbance of the mixed solution was measured at 725 nm. Linear regressions were determined using Graphpad Prism 4 software. Gallic acid was used as standard and TPC value for each food was calculated in comparison with it as GAE (mg of gallic acid equivalent to the polyphenols present in l g of dry food material).

### 4.6. DPPH Analysis

Food extracts were analyzed to measure the antioxidant capacity following the DPPH assay protocol already published [36]. In brief, a DPPH solution was prepared and diluted to 0.75 μM concentration (absorbance 0.500 ± 0.010 at 517 nm). Food extracts were diluted and analyzed at four different final concentrations ranging from 1.0 to 0.1 mg/mL. A total of 50 μL of each sample solution was added to 0.950 mL of DPPH solution and left in the dark for 30 min. Then, the absorbance of samples was measured (Shimadzu UV-2401 PC spectrophotometer). A total of 50 μL of pure methanol was used for the blank. Four measurements were recorded for each concentration. Standard deviation was always below 5%. The EC50 values were extrapolated from each graph as the concentration of sample that halves DPPH radical absorbance.

### 4.7. Statistical Analyses

In TPC and DPPH analyses, linear regressions were determined using Graphpad Prism 4 software. Statistical analyses were performed on all the data applying Student’s test and ANOVA as analysis of variance. All data resulted with *p* < 0.05 or lower.

The post run analyses were performed using Bonferroni test. All the data with *p* < 0.05 or lower were considered significantly different.

## 5. Conclusions

Cooking with a direct solar oven has proven to be not only a viable and ecological alternative to traditional baking but also a way to maintain or, in some cases, enhance the bioavailability and nutraceutical properties of the cooked vegetables and meats.

This research demonstrated that tocopherols are unaffected by solar cooking, while the total phenolic contents is stable but increased significantly in carrots, eggplants, and peppers. Solar baking also maintains or enhances (pepper, zucchini, and chicken) the antioxidant capacity of foods.

However, a loss in carotenoids and a partial isomerization of β-carotene seem to be favored by direct irradiation on the samples. The use of solar oven equipped with a UV filter has proved beneficial to avoid the drawbacks of the more energetic radiations, without losing the advantageous effects of the others.

## Figures and Tables

**Figure 1 molecules-28-03519-f001:**
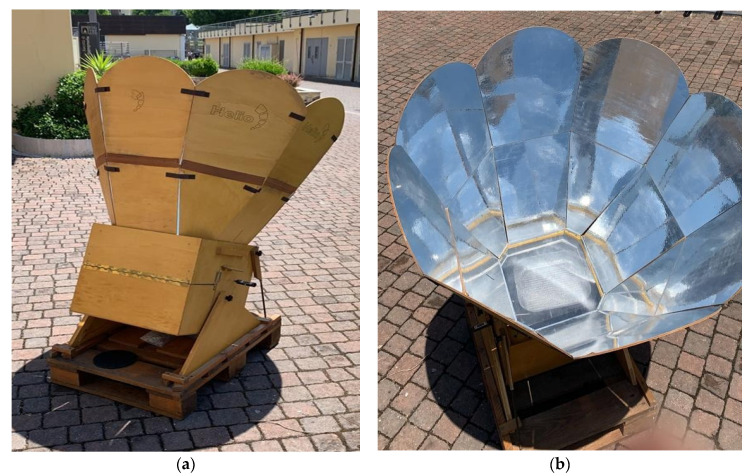
Helio© solar cooker, kindly provided by Eng. Alessandro Varesano. (**a**) Structure of the reflection panels and cooking chamber; (**b**) Vision of the cooking chamber from the glass panel. The grilling pan is visible inside.

**Figure 2 molecules-28-03519-f002:**
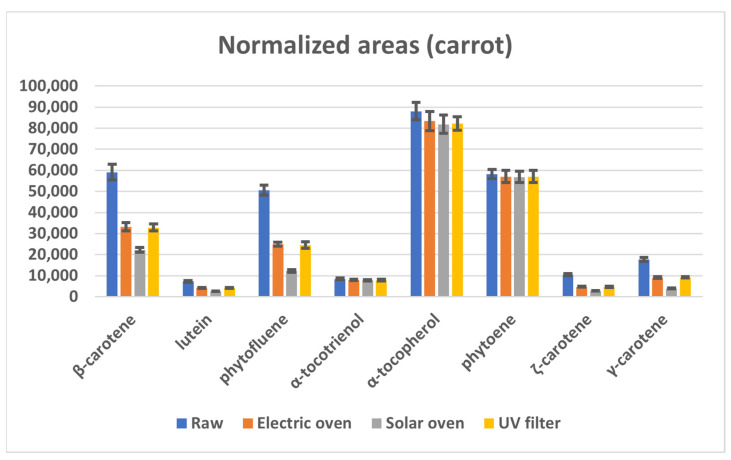
Mean chromatographic areas normalized for the weight of the sample, obtained from the HPLC-MS analyses of raw carrot samples (blue series) and after cooking by electric oven (orange series), solar oven (grey series), and solar oven with UV filter (yellow series).

**Figure 3 molecules-28-03519-f003:**
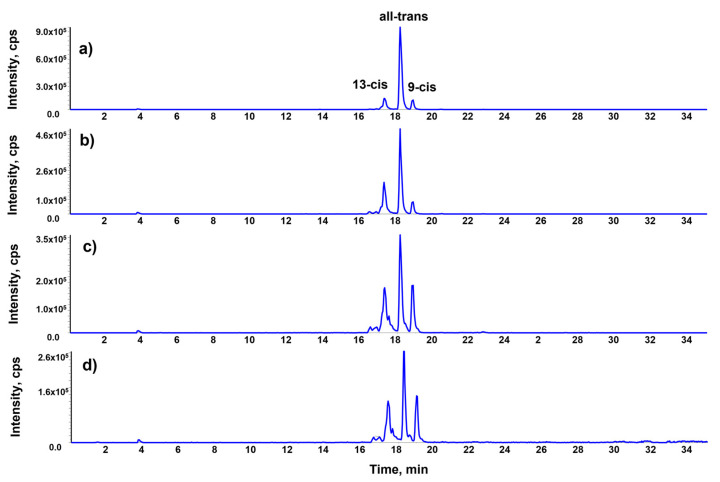
LC-MRM profiles of β-carotene in pepper: raw (**a**), cooked with electric oven (**b**), cooked with solar oven (**c**), cooked with UV filtered solar oven (**d**). y axis values are expressed as counts per second (cps).

**Table 1 molecules-28-03519-t001:** Mean chromatographic areas (×10^3^) with associated standard deviations of the fat-soluble micronutrients detected in raw and cooked foods (average values calculated on 3 replicated analyses and normalized with respect to the weight of samples).

Sample	Compound	Raw (R)	Electric Oven (E)	Solar Oven (S)	Solar Oven with UV Filter (F)
**Carrots**	β-carotene	59,120 ± 3648 ^a^	33,162 ± 2002 ^b^	22,220 ± 1119 ^c^	32,850 ± 1688 ^b^
	γ-carotene	17,630 ± 918 ^a^	9044 ± 388 ^b^	3856 ± 205 ^c^	9200 ± 358 ^b^
	ζ-carotene	10,496 ± 518 ^a^	4680 ± 188 ^b^	2830 ± 158 ^c^	4638 ± 273 ^b^
	lutein	7240 ± 449 ^a^	4176 ± 219 ^b^	2548 ± 148 ^c^	4152 ± 148 ^b^
	β-criptoxanthin	678 ± 40 ^a^	382 ± 28 ^b^	210 ± 16 ^c^	404 ± 27 ^d^
	phytoene	58,180 ± 2150	57,000 ± 2919	56,798 ± 2658	56,992 ± 2901
	phytofluene	50,560 ± 2358 ^a^	24,988 ± 960 ^b^	12,250 ± 629 ^c^	24,584 ± 1552 ^b^
	α-tocopherol	88,080 ± 4200	83,340 ± 4511	81,880 ± 4301	82,180 ± 3180
	γ-tocopherol	926 ± 51	874 ± 54	830 ± 39	840 ± 41
	α-tocotrienol	8452 ± 451	7944 ± 352	7780 ± 328	7832 ± 405
**Eggplants**	β-carotene	200 ± 12 ^a^	110 ± 6 ^b^	74 ± 4 ^c^	114 ± 6 ^d^
	lutein	48 ± 4 ^a^	28 ± 2 ^b^	18 ± 1 ^c^	26 ± 2 ^d^
	α-tocopherol	8176 ± 440	8114 ± 511	8084 ± 451	8100 ± 480
	γ-tocopherol	324 ± 21	302 ± 14	286 ± 15	294 ± 19
	α-tocotrienol	50 ± 3	44 ± 2	40 ± 2	46 ± 2
**Peppers**	β-carotene	10,396 ± 548 ^a^	5168 ± 332 ^b^	3987 ± 250 ^c^	5098 ± 230 ^b^
	lutein	1507 ± 69 ^a^	806 ± 40 ^b^	478 ± 27 ^c^	802 ± 48 ^b^
	β-criptoxanthin	2289 ± 126 ^a^	1332 ± 100 ^b^	1000 ± 53 ^c^	1326 ± 96 ^b^
	phytoene	14,344 ± 550	14,000 ± 500	13,770 ± 420	14,248 ± 501
	phytofluene	925 ± 100 ^a^	538 ± 60 ^b^	267 ± 29 ^c^	518 ± 50 ^b^
	α-tocopherol	113,080 ± 5000	1 × 10^5^ ± 4503	112,690 ± 4617	114,000 ± 5049
	γ-tocopherol	1378 ± 141	1400 ± 134	1426 ± 155	1360 ± 114
	α-tocotrienol	2640 ± 181	2515 ± 202	2603 ± 190	2482 ± 245
**Zucchini**	β-carotene	2202 ± 118 ^a^	1138 ± 52 ^b^	774 ± 39 ^c^	1142 ± 58 ^b^
	lutein	16,812 ± 949 ^a^	8632 ± 609 ^b^	4614 ± 297 ^c^	8590 ± 548 ^b^
	α-tocopherol	65,340 ± 4200	62,900 ± 3511	61,994 ± 3301	62,024 ± 3180
	γ-tocopherol	7152 ± 391	7126 ± 434	7086 ± 415	7082 ± 329
**Chicken**	β-carotene	n.d.	n.d.	n.d.	n.d.
	lutein	200 ± 12 ^a^	128 ± 9 ^b^	50 ± 3 ^c^	132 ± 8 ^b^
	phytoene	90 ± 5	86 ± 4	80 ± 4	84 ± 5
	γ-tocopherol	418 ± 31	380 ± 34	374 ± 25	388 ± 29
	α-tocotrienol	68 ± 7	60 ± 5	56 ± 4	58 ± 4
	retinol	1120 ± 65 ^a^	648 ± 34 ^b^	180 ± 12 ^c^	518 ± 31 ^d^

^a–d^ The significantly different data in each row are superscripted with different letters (Bonferroni test, *p* < 0.05); the lack of letters in a row shows that data are not significantly different (Bonferroni test, *p* > 0.05).

**Table 2 molecules-28-03519-t002:** Total phenolic content calculated as mg of gallic acid equivalent to 1 g of dry weight (mg GAE/g DW) of foods raw or cooked with different baking methods.

Sample	Raw (R)	Electric Oven (E)	Solar Oven (S)	Solar Oven with Filter (F)
carrots	20 ± 2 ^a^	19 ± 1 ^a^	22 ± 1 ^b^	26 ± 1 ^c^
onions	67 ± 8 ^a^	76 ± 10 ^b^	74 ± 10 ^b^	70 ± 8 ^a^
eggplants	146 ± 8 ^a^	150 ± 5 ^a^	207 ± 7 ^b^	187 ± 11 ^c^
peppers	159 ± 10 ^a^	163 ± 8 ^a^	180 ± 8 ^b^	159 ± 10 ^a^
zucchini	154 ± 8 ^a^	174 ± 8 ^b^	163 ± 10 ^b^	170 ± 10 ^b^
chicken	47 ± 3 ^a^	47 ± 3 ^a^	44 ± 2 ^b^	50 ± 3 ^a^
pork loin	14 ± 1 ^a^	13 ± 1 ^a^	18 ± 1 ^b^	16 ± 1 ^b^
cod fish	25 ± 1 ^a^	37 ± 1 ^b^	33 ± 2 ^c^	35 ± 1 ^c^

^a–d^ The significantly different data in each row are superscripted with different letters (Bonferroni test, *p* < 0.05).

**Table 3 molecules-28-03519-t003:** EC50 (mg/mL) of foods raw (R) or cooked with different baking methods: electric oven (E); solar oven (S) or solar oven with UV filter (F). All data derived by the determination of the DPPH scavenging activity of each food sample at, at least, four concentrations. At each concentration, antioxidant capacity was measured at least 4 times. (*p* < 0.05).

Sample	Raw (R)	Electric Oven (E)	Solar Oven (S)	Solar Oven with Filter (F)
Carrots	8.8 ± 0.3 ^a^	6.0 ± 0.2 ^b^	6.20 ± 0.09 ^b^	5.2 ± 0.1 ^c^
onions	2.7 ± 0.6 ^a^	1.3 ± 0.3 ^b^	1.4 ± 0.2 ^b^	2.7 ± 0.2 ^a^
eggplants	1.02 ± 0.02 ^a^	0.49 ± 0.01 ^b^	0.48 ± 0.08 ^b^	0.49 ± 0.01 ^b^
peppers	1.32 ± 0.1 ^a^	0.96 ± 0.02 ^b^	0.8 ± 0.1 ^c^	0.8 ± 0.1 ^c^
zucchini	5.3 ± 0.6 ^a^	2.5 ± 0.2 ^b^	1.7 ± 0.1 ^c^	2.38 ± 0.06 ^d^
chicken	33.3 ± 0.2 ^a^	27.5 ± 0.6 ^b^	25.4 ± 0.9 ^c^	24 ± 1 ^c^
pork loin	16.3 ± 0.9 ^a^	14.0 ± 0.6 ^b^	13.1 ± 0.7 ^b^	11.2 ± 0.4 ^c^
cod fish	16.5 ± 0.8 ^a^	15.9 ± 0.3 ^a^	12.6 ± 0.4 ^b^	12.5 ± 0.6 ^b^

^a–d^ The significantly different data in each row are superscripted with different letters (Bonferroni test, *p* < 0.05).

## Data Availability

Data is contained within the article.

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
