# Peer review of "Direct Solar Oven with and without UV Filter vs. Traditional Oven: Effect on Polyphenolic Antioxidants, Vitamins and Carotenoids of Food"

_molecules, 2023, doi:10.3390/molecules28083519_

Round 1

Reviewer 1 Report

ABSTRACT

Please re-write the following sentence "In the present research work, to investigate this issue, the variation of total phenolic content (TPC), lipophilic vitamins, carotenoids and antioxidant capacity of several vegetables, meats and a fish sample were analyzed before and after cooking in a traditional oven, in a solar oven and in a solar oven equipped with a UV filter."

INTRODUCTION

In the second line of the first paragraph, change the word "touches" for another one.

Make clear the following sentence "In the food industry, solar energy begins to be used to lower production costs, dry food or applied to waste transformations [2–4]."

What does mean "ecological sensitivity"?

Rewrite the following sentence "Obtaining these goals calls for high heating efficiency and tight insulation which permit to come by and maintain the temperatures similar to traditional ovens"

RESULTS

Section 2.1 gives no relevant information of the results, but only a simplified methodology. remove it

the same applies to the first paragraph of section 2.2. this one could fit better in discussion

IF the authors used PRISM for the statistical analysis, please also use it for the creation of figure 2.

MATERIALS AND METHODS

Since Molecules uses a format different to most journals, is easy to get lost. Please include the HELIO sun dryer in this section rather than in RESULTS (Section 2).

All scientific names must be written using italics

CONCLUSIONS

There are no conclusions, please add it

Reviewer 2 Report

As the attached PDF file

Round 2

Reviewer 2 Report

The authors have addressed all the concerns raised in my previous review and have made the necessary changes to the manuscript. The manuscript is more clear and concise and I believe that the manuscript can be accepted for publication in its present form.